# Obeying the Order: Introducing
# Ordered Transfer Hyperparameter Optimization

**Sigrid Passano Hellan**[*, 1,‡]  **Huibin Shen**[2]  **François-Xavier Aubet**[2]  **David Salinas**[3,4,†]
**Aaron Klein**[3,5,†]

[1]NORCE Research and Bjerknes Centre for Climate Research, Bergen, Norway
[2]Amazon Web Services (AWS)
[3]ELLIS Institute Tübingen
[4]University of Freiburg
[5]ScaDS.AI Leipzig
[†]Work started while at AWS
[‡]Work started while interning at AWS

**Abstract**  In many deployed settings, hyperparameters are retuned as more data are collected; for instance tuning a sequence of movie recommendation systems as more movies and rating are added. Despite this, transfer hyperparameter optimisation (HPO) has not been thoroughly analysed in this setting. We introduce *ordered transfer hyperparameter optimisation* (OTHPO), a version of transfer learning for HPO where the tasks follow a sequential order. Unlike for state-of-the-art transfer HPO, the assumption is that each task is most correlated to those immediately before it. We propose a formal definition and illustrate the key difference with standard transfer HPO approaches. We show how simple methods taking the order into account can outperform more sophisticated transfer methods by better tracking smooth shifts of the hyperparameter landscape. The ten benchmarks are in the setting of gradually accumulating data, as well as a separate real-world motivated optimisation problem, and are open sourced to foster future research on ordered transfer HPO.

## 1 Introduction

All modern machine learning (ML) pipelines contain many hyperparameters that are critical for final performance, as they govern key parts of the training such as the optimisation (learning rate), the capacity of the model (number of layers or regularisation weights) or data augmentation. Hyperparameter optimisation (HPO) — see e.g. Feurer and Hutter (2019) — aims to find the optimal hyperparameters of a machine learning method by casting it as an optimisation problem: for each iteration a new set of hyperparameters is used to train and validate the method.

In practical scenarios, hyperparameters are not tuned once but many times. Consider a movie recommendation ML system being deployed. The model hyperparameters must be tuned frequently, given that the data set consistently evolves with new movies and users, which increases the data set size and gives a continuous shift in the optimal hyperparameter values. In particular, we expect some hyperparameters that define the regularisation or the capacity of the model to change as more data is observed. Smaller models might be initially superior for little data, as only simple rules can be learned, but, as more data becomes available, more expressive models start to become competitive, as they are able to identify smaller differences between the inputs. This point is illustrated in Fig. 1 which plots the validation performance of XGBoost with respect to the number of estimators and the maximum depth of each tree. As the data set size increases (from left to right) the optimal hyperparameter values change smoothly and more expressive models (i.e. more and deeper trees) become superior. We call each hyperparameter optimisation of a model a *task*.

---

[*]Correspondence to: Sigrid Passano Hellan <sipa@norceresearch.no>.

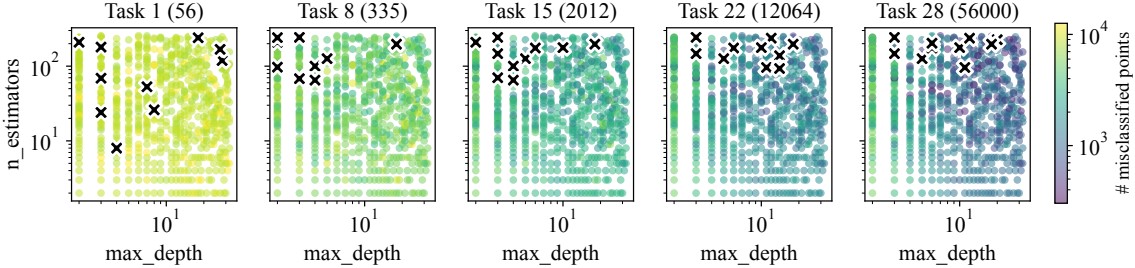

Figure 1: Evolution of the optimal XGBoost hyperparameters on MNIST for gradually increasing training set sizes (from 56 to 56000 points). The black crosses indicate the 10 best hyperparameters, and can be seen to shift upwards to more estimators as more data is added.

A popular family of approaches that can exploit information from previous tasks is transfer HPO (Wistuba et al., 2015b; Feurer et al., 2018; Salinas et al., 2020). Such methods exploit data collected from the HPO of previous tasks to warm-start the optimisation on the current task. However, transfer HPO methods treat tasks as a set and ignore any intrinsic order. In practice, data is often collected in a sequence, for instance when a production system is tuned at frequent intervals. We introduce *ordered transfer HPO* (OTHPO), as a special case of transfer HPO that exploits this sequential nature of tasks, enabling a better transfer of knowledge across tasks. See Fig. 2 for an illustration. Our contributions are:

- We propose a formal definition of ordered transfer HPO and outline the differences to related problems, such as standard transfer HPO and continual learning.
- We provide ten benchmarks for this setup, integrated in the open-source HPO library Syne Tune (Salinas et al., 2022) to compare existing approaches for transfer HPO and foster the development of future methods. These benchmarks include XGBoost (Chen and Guestrin, 2016), support vector machines (Schölkopf and Smola, 2002), approximate k-nearest neighbor (Malkov and Yashunin, 2018), random forest (Wright and Ziegler, 2017) and elastic net (Friedman et al., 2010) on various data sets, as well as a blackbox optimisation task based on SimOpt (Eckman et al., 2023).
- Our results in the setting of accumulating training data over time suggest that, in this setting, OTHPO methods taking order into account are simple and performant, which provides guidance for HPO practitioners in a deployed system.

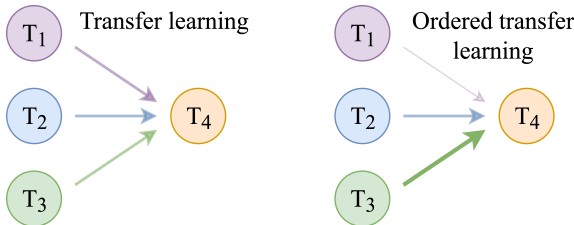

Figure 2: Contrasting standard transfer HPO and OTHPO. In the former (left), when learning the next task, $T_4$, each previous task ($T_1$, $T_2$, $T_3$) is either assumed equally important or weighted according to meta-features or hyperparameter rank matching. For OTHPO (right), more recent tasks are assumed more relevant, as illustrated by the difference in arrow widths.

## 2 Related work

OTHPO is related to but distinct from standard transfer HPO (Bai et al., 2023), continual learning (Van de Ven and Tolias, 2019; Chaudhry et al., 2019) and multi-fidelity HPO (Jamieson and Talwalkar, 2016; Li et al., 2017).

**Standard transfer HPO** (Perrone et al., 2019; Salinas et al., 2020; Wistuba et al., 2015b; Horváth et al., 2021) typically transfers knowledge between data sets, and, compared to OTHPO tasks, does not have an inherent ordering that one can exploit. For example, Wistuba et al. (2015b) used hyperparameter configurations evaluated on previous tasks to create a portfolio of well-performing configurations, which are sequentially evaluated on a new task. Perrone et al. (2019) reduced the search space for a new task by defining a bounding box around the best performing configurations on all previous tasks. To account for the variability in objective scales of different tasks, Salinas et al. (2020) learned a semi-parametric Gaussian Copula distribution across tasks. Springenberg et al. (2016) used a Bayesian neural network to model the correlation between tasks for multi-task Bayesian optimisation. In a similar vein, Perrone et al. (2018) trained neural networks to learn basis functions across tasks and combined this with Bayesian linear regression to obtain reliable uncertainty estimates for Bayesian optimisation. This idea was extended by Horváth et al. (2021), which regularised the basis functions to account for the changing complexity during the optimisation process.

In Wistuba and Grabocka (2021), the authors considered HPO as a few-shot learning problem where a Deep GP (Gaussian process) model was trained jointly on a set of meta-tasks by few-shot learning. For a target task they then started from the initialised kernel parameters, before fine-tuning the model with a few hyperparameter evaluations.

Some transfer learning HPO methods such as Wistuba et al. (2015a) proposed leveraging meta-features of previous tasks to exploit task similarities. However, those methods rely on manual engineering of task features which are critical for final performance. To tackle this issue, Jomaa et al. (2021) proposed using Deep GPs to learn meta-features in an end-to-end fashion, which shows encouraging results in combating negative transfers. In what follows, we use standard transfer HPO as a shorthand for non-ordered transfer learning.

Both OTHPO and **continual learning** consider sequences of tasks. But continual learning is concerned with maintaining model performance on previous tasks, typically keeping the hyperparameters constant (Lee et al., 2024). We, on the other hand, want to optimise the hyperparameters for the current task and do not need the new model to perform well on the previous tasks.

In contrast to **multi-fidelity HPO**, OTHPO cares about the performance at each level. While a subset of the training data could be used in the multi-fidelity setting as a heuristic for later performance (Li et al., 2017; Klein et al., 2017), we consider the performance on the earlier task as a goal in itself. The idea (Zappella et al., 2021) of multi-fidelity HPO has also been extended to the standard transfer HPO setting.

Previous work has also been motivated by the idea of considering HPO of a model under change as a sequence of tasks (Golovin et al., 2017; Zhang et al., 2019; Stoll et al., 2020), but to our knowledge none have explicitly evaluated the importance of the ordering and they exhibit a few notable differences to our work. In Stoll et al. (2020), the ordering is implicitly used by only transferring from the latest task with a different search space and underlying algorithm. Golovin et al. (2017) motivate their work through HPO on different tasks in a sequence, but do not present results on HPO or compare to standard transfer HPO. The open source version of the software does not include transfer learning, so is not included as a baseline in this paper.

Zhang et al. (2019) is most related to our work as they identify the practical problem of slowly evolving data sets and the need to perform transfer HPO. However, they do not explicitly compare using the task ordering to standard transfer HPO. And while their evaluation considers the best possible performance on a new task, we show the large potential speed-ups possible by using OTHPO, since we show improved results after only one hyperparameter evaluation. The OTHPO method we propose is much simpler than those in Golovin et al. (2017); Zhang et al. (2019). This means we can directly evaluate the benefit of taking the ordering into account.

Before going further, we want to remind our readers that our goal is not to propose another transfer HPO method but rather to introduce a setting that is relevant for practitioners using HPO in a deployed system, and demonstrate the potential of utilising the sequential nature of the tasks.

## 3 Problem definition

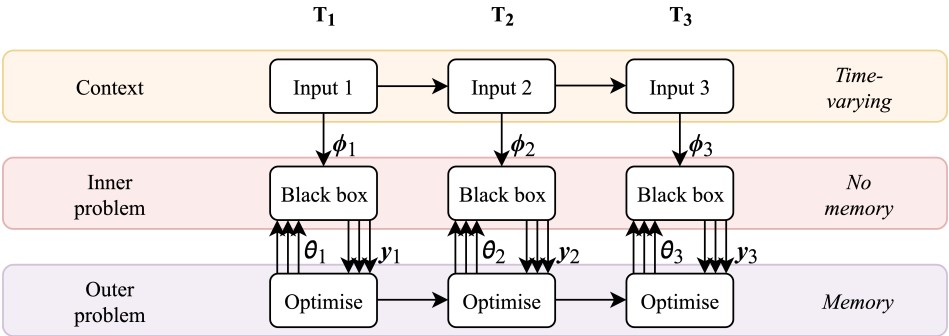

Figure 3: Structure of an ordered transfer HPO problem. The tasks $T_1$, $T_2$, $T_3$ have an inherent order in the outer problem stemming from the drift in the context. There is no connection between the inner problems. For the notation see Section 3.

Let $f : \Theta \to \mathbb{R}$ denote the validation performance of a machine learning algorithm after training with hyperparameters $\boldsymbol{\theta} \in \Theta$. HPO treats the search for the optimal hyperparameters as a global optimisation problem $\boldsymbol{\theta}^* = \operatorname*{argmin}_{\boldsymbol{\theta} \in \Theta} f(\boldsymbol{\theta})$. The space of all possible hyperparameter configurations $\boldsymbol{\theta} \in \Theta$ is called the configuration space. Due to the intrinsic randomness of most machine learning methods, for example random weight initialisation or mini-batch sampling, we observe $f$ only with noise: $\boldsymbol{y} = f(\boldsymbol{\theta}) + \epsilon$, where $\epsilon \sim \mathcal{N}(0, \sigma^2)$.

In practice, we often face the same HPO problem repeatedly on different tasks, where the configuration space and the underlying machine learning algorithm are the same, but training (and possibly validation) data sets change. To share knowledge across HPO tasks, we treat the objective functions as a series of related global optimisation problems, see Fig. 3. Here the outer problem could be HPO, the inner problem could be training an ML model, and the context could be the training data. More formally, we augment the definition of our objective function $f : \Theta \times \mathbb{V} \to \mathbb{R}$ by another input $\boldsymbol{\phi}_i \in \mathbb{V}$ that denotes the current task $i$. Now, OTHPO assumes that tasks come in a sequence, such that task $i$ is more similar to task $i - 1$ than to task $i - 2$.

For task $i$, we collect evaluations $\boldsymbol{y}_{i,m} = f(\boldsymbol{\theta}_{i,m}; \boldsymbol{\phi}_i) + \epsilon$ through the optimiser in the outer problem. The subscript $m$ denotes which of the $M_i$ evaluations of the inner problem $i$ is given. When deciding what configuration to try next, the outer optimiser has access to the evaluations of previous tasks $\{\boldsymbol{\theta}_{j,m}, \boldsymbol{y}_{j,m}\}_{j=1, m=1}^{i-1, M_j}$ and all the $N$ finished evaluations in the current task $\{\boldsymbol{\theta}_{i,m}, \boldsymbol{y}_{i,m}\}_{m=1}^{N}$. We assume that the search space $\Theta$ remains the same between tasks.

## 4 Benchmarks

We propose 10 benchmarks to evaluate methods on OTHPO, summarised in Table 1. We implement our benchmarks in Syne Tune (Salinas et al., 2022), making them easily available to everyone. In this section we describe each benchmark in more depth.

**XGBoost on MNIST.** In a deployed setting, one collects more training data as time passes. When refitting the model on more data it is likely that the optimal hyperparameters on earlier data sets are not optimal anymore. We propose an evaluation benchmark for this setting by training an XGBoost classifier (Chen and Guestrin, 2016) on the MNIST data set (Vanschoren et al., 2013) with increasing

|  | XGBoost | YAHPO | NewsVendor |
|---|---|---|---|
| Context | Training data size | Training data size | Environment settings |
| Inner problem | Minimise error | Maximise AUC | Simulate profit |
| Outer problem | HPO | HPO | Parameter optimisation |
| Number of tasks | 28 | 20 | 9 |
| Number of benchmarks | 1 | 8 | 1 |

Table 1: Overview of benchmarks, using concepts from Fig. 3.

training set sizes. We tune four hyperparameters: max_depth, n_estimators, min_child_weight and learning_rate . We have 28 tasks, with training set sizes regularly selected in the log space, ranging from 56 to 56000 training examples (see Appendix B). Some of these tasks are shown in Fig. 1. We use surrogate models fit on 1000 hyperparameter evaluations to avoid any model training during HPO. We ensure all ten classes are represented in every task. Our optimisation metric is the number of misclassified points in a validation set comprising 14000 examples.

**YAHPO.** The next 8 benchmarks are drawn from YAHPO Gym (Pfisterer et al., 2022), a HPO benchmark suite containing a variety of HPO problems. We focus on a subset of scenarios from RandomRobot version 2 (rbv2) because they have different training set sizes available. Our initial investigation suggests that, depending on the ML model and the data set, the top performing hyperparameters are either smoothly changing over increasing training set size or stay in a similar region. We select smoothly changing ones for our benchmarks. An example is given in Fig. 4 (top), for more see Appendix F.2. In YAHPO, surrogate models are also used to predict hyperparameter performances for faster experimentation.

We consider four diverse ML models with two data sets each, so eight data sets in total, and optimise the AUC. The models considered (Binder et al., 2020) are SVM (support vector machines, Schölkopf and Smola (2002)), AKNN (approximate k-nearest neighbor, Malkov and Yashunin (2018)), ranger (random forest, Wright and Ziegler (2017)) and glmnet (elastic net, Friedman et al. (2010)). The data set ID will be shown next to the algorithm, e.g. SVM 1220. We create 20 tasks by gradually increasing the size of the training data set from 5% to 100 %.

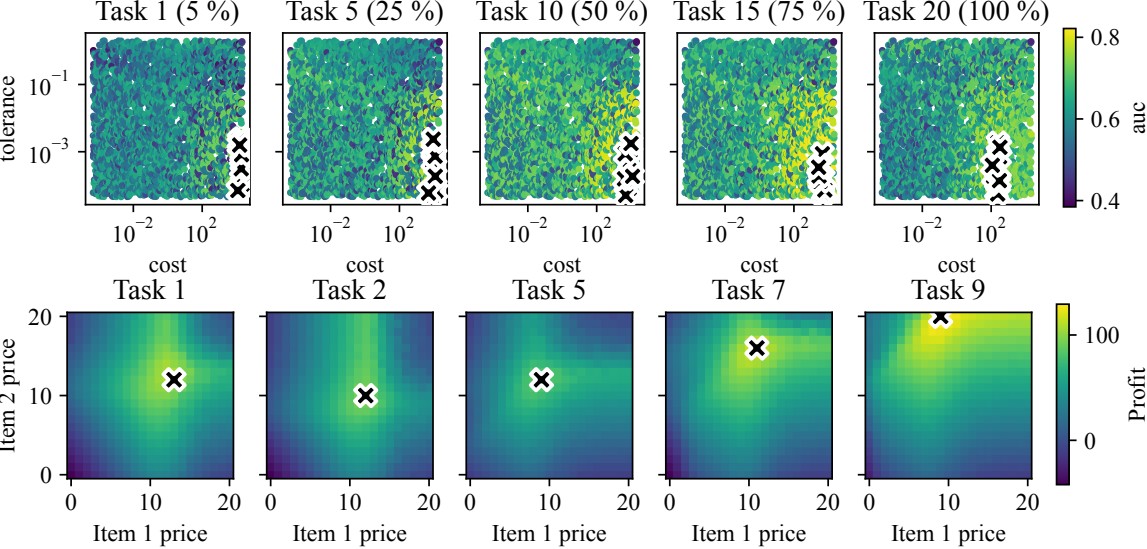

Figure 4: Hyperparameter landscapes of YAHPO SVM 1220 (top) and NewsVendor (bottom). Crosses indicate the 10 best configurations per task for YAHPO and the best one for NewsVendor.

**NewsVendor.** The benchmarks presented so far are examples of HPO as data set sizes increase. However, many other kinds of systems need to be optimised periodically due to evolving environmental conditions. In NewsVendor, the aim is to maximise profit by setting the prices for three item categories given the current uncertain item demands. This benchmark is based on the Dynamic News problem from the SimOpt library (Eckman et al., 2023). Over time the item demand, or utility, is influenced by external factors and evolves. Here we simulate the change by following a random walk on the utility, resulting in the sequence of tasks. Note that this means the context is not necessarily changing in a single direction as for XGBoost and YAHPO. This is illustrated in Fig. 4 (bottom). We consider a sequence of 9 tasks, with three item categories, making the outer problem 3-dimensional with an integer search space limited between 0 and 20.

## 5 Experiments

To show the potential gain available in transfer HPO from taking the task order into account, we compare simple OTHPO methods to non-transfer Bayesian optimisation and standard transfer HPO methods. Our code is available at `https://github.com/sighellan/syne-tune/tree/othpo-results`.

### 5.1 Baselines

We consider several non-transfer and transfer HPO methods from the literature as baselines:

- **RandomSearch**: Sample configurations uniformly at random from the search space.
- **BO** (Snoek et al., 2012): Run Bayesian optimisation with no transfer between tasks.
- **BoundingBox** (Perrone et al., 2019): Shrink the search space of BO to the bounding box of optima on previous tasks. Note that this means the search space cannot increase for future tasks and it requires two finished tasks so a box can be computed.
- **ZeroShot** (Wistuba et al., 2015b): Learn a portfolio of complementary hyperparameters with greedy selection based on previous tasks performance. The configurations of the portfolio are then evaluated sequentially.
- **CTS** (Copula Thompson Sampling, Salinas et al. (2020)): Map the evaluations to quantiles within each task and learn a probabilistic model to predict the quantiles. For a new task, the method then samples the performance of each candidate configuration and picks the configuration with the lowest sampled value.

We use implementations in Syne Tune for our baselines and BoTorch (Balandat et al., 2020) for BO and the transfer learning methods relying on BO. We use a Matérn 5/2 kernel and a Monte Carlo version of expected improvement (Jones et al., 1998), see Appendix E. The HPO tasks are sequentially evaluated, with the evaluations collected in one task available for the subsequent tasks.

### 5.2 Simple ordered transfer HPO methods

- **TransferBO**: *Extend the BO surrogate model to also take the task order as an input feature.* Evaluations from the current and previous tasks are used to train a GP where the task order is explicitly modelled through a task feature: For NewsVendor, we use the task index as the feature; for the other benchmarks, we use the training set size. The idea of modelling training set sizes in the surrogate model have been widely used (Klein et al., 2017, 2020). We use the simplest form where the same kernel function is applied on both hyperparameters and training set sizes. For computational reasons we only use 200 evaluations to train each GP, see Appendix E.
- **SimpleOrdered**: *Standard BO, but start by evaluating top-performing hyperparameters from the most recent tasks.* For a new task, the first $N$ hyperparameter configurations come from the top hyperparameter configurations of each of the previous $N$ tasks, starting with the most recent one and continuing in reverse order of time. Then we continue with standard BO. We set $N = 5$ in

our experiments. For edge cases see Appendix D. We also test a variant **SimpleOrderedShuffled** in ablations where we still take top configurations from previous tasks, but in random order.

- **SimplePrevious**: *Same as SimpleOrdered, but only using the last, previous task.* The initial $N = 5$ hyperparameters are the best $N$ hyperparameters from the previous task. It is a simplified version of Feurer et al. (2015) where meta-feature computation can be avoided due to our assumption that the closest task in the sequence is the most similar task. We also test a variant **SimplePreviousNoBO** which only uses hyperparameters from the previous task, sorted by decreasing performance, without any BO.

## 5.3 Experimental setup and metric

**Experimental setup**: For each benchmark described in Section 4, we sequentially apply HPO and transfer HPO methods for each tuning task with the number of hyperparameter evaluations restricted to 25. We rerun each experiment with 50 seeds and report average performance ± 2 standard errors when plotting method results. The transfer learning methods assume evaluations from at least one previous task. All these methods therefore use BO to collect evaluations on the first task. BoundingBox also uses it on the second task, see Appendix C.

**Metric**: For ease of aggregation, we use a version of the Normalised Score from (Cowen-Rivers et al., 2022, eq. 3). Let $i$ index tasks, $m$ index iterations within a HPO task and $M$ be the maximum number of hyperparameter evaluations for each task. The score is defined as $100 * (L_{i,m} - L_{i,M}^{\text{best}})/(L_{i,M}^{\text{RS}} - L_{i,M}^{\text{best}})$, where $L_{i,m}$ is the mean loss across replications for a method on task $i$ at iteration $m$, $L_{i,M}^{\text{best}}$ is the estimated best solution for the task and $L_{i,M}^{\text{RS}}$ is the mean performance of RandomSearch on the task by the final iteration. For $L_{i,M}^{\text{best}}$ we use the best mean obtained across the compared methods.

The Normalised Score computes the loss distance at an iteration to the best solution, normalised by the loss distance between RandomSearch at the final iteration $M$ and the best solution, thus the smaller the better, ideally 0. It allows easy comparison between minimisation (XGBoost) and maximisation (NewsVendor, YAHPO) benchmarks, and also normalises tasks by difficulty and away from the scale of the optimisation metric. We present results after 1 and 10 iterations in the main paper. Fig. 5 compares all ten methods on NewsVendor, XGBoost and SVM 1220 after 1 iteration. Fig. 7 compares the best OTHPO methods, SimpleOrdered and SimplePrevious, on the eight YAHPO combinations after 1 iteration. Fig. 6 compares SimpleOrdered and TransferBO to the top-performing standard transfer HPO method CTS after 1 and 10 iterations. Further results are given in Appendix G.

## 5.4 First evaluation: SimpleOrdered and SimplePrevious beat standard transfer HPO

Using ordering gives a large benefit at the first evaluation. This can be seen in Fig. 5 and Fig. 6 (top). We first note that all the transfer HPO methods, including SimpleOrdered, outperform non-transfer methods. SimpleOrdered mostly beats — or at least is on par compared to — all the baselines. For NewsVendor, SimpleOrdered does slightly worse than for the other benchmarks. This might be because the number of previous tasks that can be used is relatively small, and the changes between tasks is not a gradual shift in one direction like in the other benchmarks. We note that SimpleOrdered is a very simple method in comparison to the standard transfer HPO baseline methods. TransferBO performs worse than SimpleOrdered. This shows that TransferBO is not able to use the task index effectively. We investigate the sampling pattern in Appendix G.6.

In Fig. 5 we see that SimpleOrdered and SimplePrevious outperform standard transfer HPO. We compare SimpleOrdered and SimplePrevious in Fig. 7, and see that the performance is very similar, with SimpleOrdered slightly preferable on early tasks and SimplePrevious on later tasks. This highlights the large impact of the most recent task.

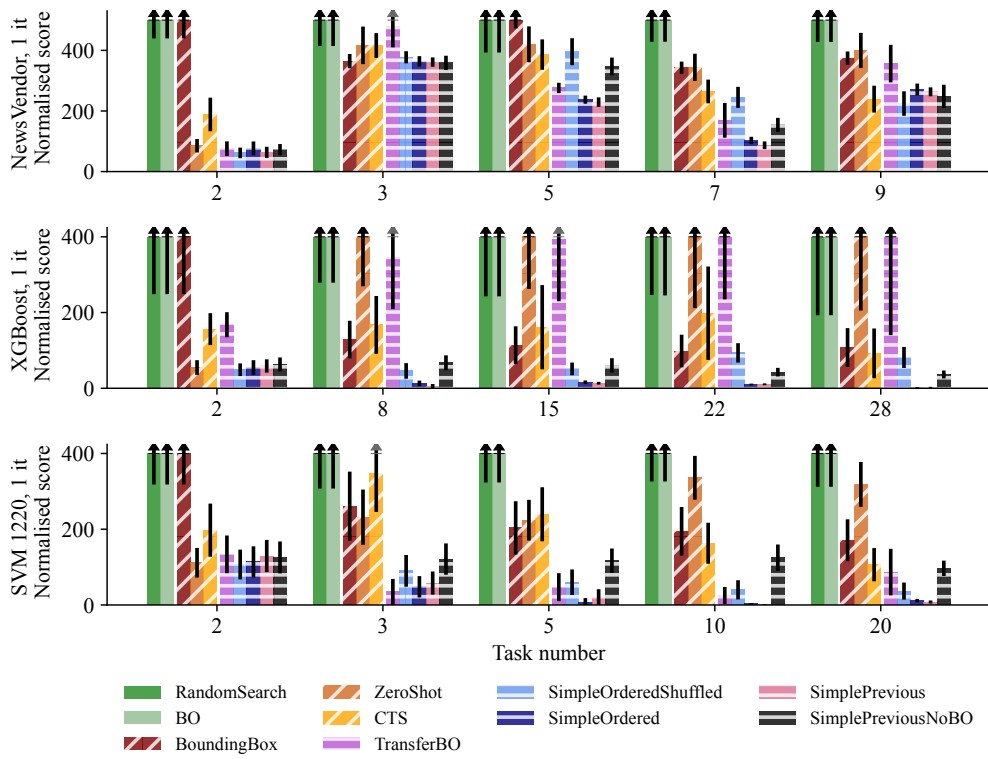

Figure 5: Mean normalised scores after first configuration (lower is better). NewsVendor (top), XGBoost (middle) and SVM 1220 (bottom). Black arrows indicate that the mean was above the plotted range, grey arrows that the standard error range was above. The methods are grouped by colour with OTHPO methods in blue, purple and pink, standard transfer HPO methods in yellow and red, and methods with no transfer in green. Task 2 is the first transfer task.

We further analyse the impact of ordering and surrogates in ablations in Appendices G.1 and G.7, which show that the lack of ordering in SimpleOrderedShuffled is detrimental, as is the lack of BO in SimplePreviousNoBO. We also compare to a different surrogate model (Tiao et al., 2021).

### 5.5 Ordered advantage reduces with more evaluations

While SimpleOrdered and SimplePrevious are clearly better after the first evaluation, this becomes benchmark-specific after the fifth evaluation. As can be seen in Fig. 6 (bottom), TransferBO is best for NewsVendor, CTS is best for XGBoost and SimpleOrdered is best for SVM 1220. We investigate this further in Appendices G.2 to G.4. It is difficult to declare a method best because the variance between runs and between benchmarks becomes too high. The general trend is that SimpleOrdered and SimplePrevious pick better first configurations, and the other methods catch up with more evaluations. But SimpleOrdered and SimplePrevious remain reliable choices even for greater numbers of evaluations. While TransferBO and CTS beat them on individual benchmarks, they do worse on other ones. From this we conclude that; a) there is no universal method to prefer after five evaluations, it depends on the set of tasks; and b) there is scope for improved methods combining SimpleOrdered with either TransferBO or CTS to come up with a stronger method. We also expect other more sophisticated OTHPO methods to be able to outperform SimpleOrdered.

### 5.6 How much better will the trained models be?

The normalised score is very useful for comparing methods across tasks and benchmarks. But it abstracts away the potential performance gain, leaving the question: *how much better will my model perform after the first configuration if I use SimpleOrdered instead of CTS?*

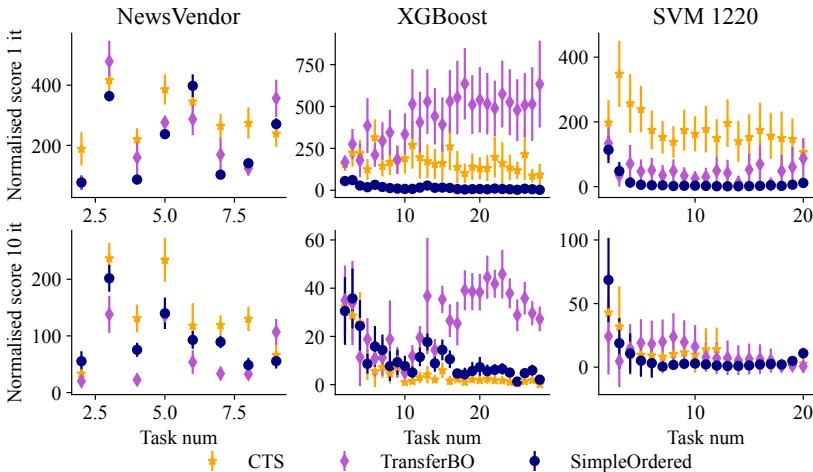

Figure 6: Mean normalised scores after first (top) and tenth (bottom) configuration. Top: SimpleOrdered clearly outperforms TransferBO and CTS. Bottom: For NewsVendor and XGBoost we see that TransferBO and CTS, respectively, are able to find better solutions than SimpleOrdered.

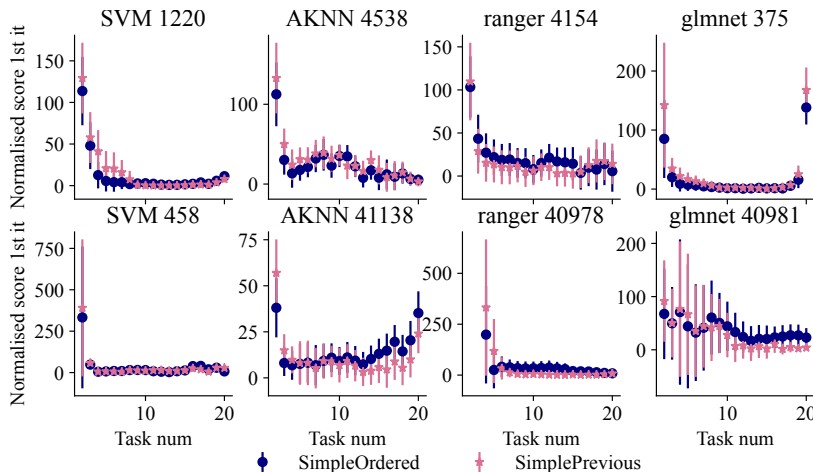

Figure 7: Mean normalised scores after first configuration evaluation for all eight YAHPO combinations considered. We see that SimpleOrdered and SimplePrevious perform very similarly.

There are two benefits: better average performance and less variance. Using the metrics of the underlying tasks, we get a mean improvement of 21.7% in profit for NewsVendor, 22.5 % in number of misclassified points in XGBoost and 5.8 % in AUC for SVM 1220. The lower variance is also very valuable, as it reduces the need to retrain models with multiple seeds to get a good model. The reduction in standard error of using SimpleOrdered instead of CTS is 61.3 % for NewsVendor, 92.5 % for XGBoost and 89.4 % for SVM 1220. We summarise these numbers in a table in Appendix G.5, where we also give error bounds. The higher variance of CTS is also visible in Figs. 5 and 6.

## 6   Conclusion

We examined the potential of ordered transfer, focussing on Bayesian optimisation methods and increasing trainings set sizes. We illustrated the key difference with standard transfer HPO approaches and showed how simple methods taking the order into account can outperform more sophisticated transfer methods by better tracking smooth shifts of the hyperparameter landscape. Due to their simplicity, the methods let us examine the benefits of ordering. We hope that our

simple methods will be useful to enable the regular tuning of deployed methods while containing tuning costs, and that the benchmarks will enable evaluating methods in this practical setting.

**Limitations**: We focused on a situation that commonly occurs for deployed models, namely that the size of the training data increases over time, but other sequences of tasks should also be explored, e.g. when the model capacity or number of classes increases over time. In addition, while we show good performance for the simple method proposed — especially for small HPO budgets — we believe further gain can be achieved with more sophisticated methods that decide for instance whether tuning is needed on the new tasks and automatically terminates in cases where tasks do not change, similarly to Makarova et al. (2022). Additionally, the principles of ordered transfer can be adopted into other HPO approaches than Bayesian optimisation, for instance to inform the starting configurations of successive halving methods (Li et al., 2020).

**Practical recommendation**: Our results show that in this setting of accumulating data, SimpleOrdered performs well, especially on early iterations for a new task. We therefore recommend practitioners to start with this simple method before trying more sophisticated ones.

**Acknowledgements**. The authors would like to thank Jan Gasthaus, Valerio Perrone, Martin Wistuba and Michael Bohlke-Schneider for help with the project.

Hellan was supported by the EPSRC Centre for Doctoral Training in Data Science, funded by the UK Engineering and Physical Sciences Research Council (grant EP/L016427/1) and the University of Edinburgh.

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

## A Assets used

We use benchmarks from three families:
- YAHPO (Pfisterer et al., 2022) (Apache License 2.0)
  - We only handle the surrogate benchmarks in YAHPO, not the data used to generate them. But we list these data sets here. For the data sets citing Dua and Graff (2017) the authors of the repository request citation.
  - Data set 1220 (Organizers of KDD Cup 2012 and Tencent Inc, 2012). Requires attribution, and the data are restricted to be used for scientific research purposes only. License: Public
  - Data set 4538. Requires citation of Dua and Graff (2017) and a relevant paper, e.g. Madeo et al. (2013) License: Public
  - Data set 4154. License: Public
  - Data set 375 (Dua and Graff, 2017; Kudo et al., 1999). Author asked to be informed about published work using the data. License: Public
  - Data set 458 (Simonoff, 2003). Requires attribution, and data are restricted to be used for scientific, educational and/or noncommercial purposes. License: Public
  - Data set 41138 (Dua and Graff, 2017). License: GNU GPL v3
  - Data set 40978 (Dua and Graff, 2017; Kushmerick, 1999). License: Public
  - Data set 40981 (Dua and Graff, 2017; Quinlan, 1987). License: Public
- NewsVendor, based on SimOpt (Eckman et al., 2023) (MIT license):
  - No underlying data set. We generate the changing utilities using a random walk.
- XGBoost (Chen and Guestrin, 2016) (Apache License 2.0)
  - Data set: mnist_784 downloaded from Vanschoren et al. (2013)

The only data we handled was the MNIST data. It does not contain any personally identifiable or offensive content.

## B XGBoost evaluations collection

For our XGBoost benchmark we collected evaluations of 1000 hyperparameter configurations on 28 training data set sizes, which we used as the basis of our simulation. The hyperparameter configurations were randomly selected and evaluated on each of the 28 data set sizes. The search space for the hyperparameters is given in Table 2 and the data set sizes in Table 3.

| Hyperparameter | Type | Min | Max | Scaling |
|---|---|---|---|---|
| learning_rate | Cont. | 1e-6 | 1 | log |
| min_child_weight | Cont. | 1e-6 | 32 | log |
| max_depth | Int | 2 | 32 | log |
| n_estimators | Int | 2 | 256 | log |

Table 2: Hyperparameter search space used for XGBoost.

| | | | | | | | | |
|---|---|---|---|---|---|---|---|---|
| 56 | 72 | 93 | 120 | 155 | 201 | 259 | 335 | 433 |
| 560 | 723 | 934 | 1206 | 1558 | 2012 | 2599 | 3357 | 4335 |
| 5600 | 7232 | 9341 | 12064 | 15582 | 20125 | 25992 | 33571 | 43358 |
| 56000 | | | | | | | | |

Table 3: Data set sizes evaluated for XGBoost.

## C  Additional details on the experimental setup

**Seeds**: We rerun each method on each benchmark 50 times. We use integers between 0 and 49 as seeds.

The transfer learning methods require evaluations from previous tasks. We use BO to collect these. For all methods except BoundingBox this was done for one task (task 1). For BoundingBox we do two tasks (tasks 1 and 2), as otherwise the method collapses to only trying the best evaluation from the first task on any of the future tasks. More specifically, we run BO until we have different optima for at least two tasks, as otherwise we also get the situation of the bounding box only containing one configuration.

## D  SimpleOrdered implementation details

The first $N$ configurations are used to evaluate the top configuration from each of the previous $N$ tasks. We do this in reverse order, i.e. at task $i$ we first evaluate the configuration from task $i - 1$, then from task $i - 2$ and so on until task $i - N$.

There are several situations which require modifications to this:

- There are $L < N$ previous tasks: in this case, we pick the top configuration from each of the $L$ tasks, and then pick the second-best configuration from each of the tasks, until we reach $N$ configurations. If we still don't have $N$ configurations we continue with the third-best, and so on.
- There are repeated optimal configurations: in this case the configuration is skipped. That means that we might end up using the top configuration from task $i - (N + 1)$.
- There are joint optima: this is the situation if two hyperparameter configurations both perform best. We attempt to pick the first of these configurations, but if it is a repeat it will be skipped. We then add the other joint optima to the back of the list we are considering. So if there are $L < N$ previous tasks it might get selected once we have included one optima from each of the $L$ tasks.
    The code for SimpleOrdered is available together with the rest of the paper code.

## E  BO details

We use a slightly updated version of the BoTorch-based (Balandat et al., 2020) BO implementation in Syne Tune (Salinas et al., 2022), see `https://github.com/sighellan/syne-tune/blob/othpo-results/syne_tune/optimizer/schedulers/searchers/botorch/botorch_searcher.py`. The acquisition function is a Monte Carlo version of expected improvement (Jones et al., 1998), and the covariance function Matérn 5/2. We also apply input warping.

For TransferBO the maximum number of observations is set to 200. For later tasks, we therefore subsample 200 of the past/current samples.

## F  Additional hyperparameter landscapes

### F.1  XGBoost

Fig. 8 compares the top hyperparameters for the first and last tasks of the XGBoost benchmark. We show all six possible 2-dimensional combinations of the four hyperparameters.

### F.2  YAHPO

Figs. 9 and 10 show additional YAHPO hyperparameter landscapes for the model – data set combinations used in Fig. 7. The final hyperparameter landscape used is shown in Fig. 4 (top).

## G  Additional results

### G.1  Ablations

We perform two ablation experiments. In Fig. 11 we compare SimpleOrdered to SimpleOrdered-Shuffled, the version that takes top points from randomly chosen previous tasks. As can be seen, the ordered version does much better.

|  | NewsVendor | XGBoost | SVM 1220 |
|---|---|---|---|
| Reduction in standard error SimpleOrdered over CTS (%) | 61.3 $\quad(53.7-68.9)$ | 92.5 $\quad(88.9-96.0)$ | 89.4 $\quad(83.0-95.9)$ |
| Improvement in mean SimpleOrdered over CTS (%) | 21.7 $\quad(8.0-35.4)$ | 22.5 $\quad(18.2-26.7)$ | 5.8 $\quad(4.9-6.7)$ |

Table 4: Downstream comparison of SimpleOrdered and CTS after one configuration. SimpleOrdered gives an improvement on all benchmarks, and the standard error is smaller, making it more reliable. We present means +- two standard errors for each value.

In Fig. 12 we compare SimplePrevious to SimplePreviousNoBO, the version that only considers configurations used in the previous task. That means that throughout the tasks only the configurations used by BO in the initial task are considered. As can be seen, the version that includes exploration through BO does much better.

### G.2 Rankings over evaluations

Fig. 13 shows the mean internal rankings among the ten methods for a subset of the methods. We see that SimpleOrdered and SimplePrevious do very well at the beginning, and are then overtaken by either TransferBO or CTS.

### G.3 Additional bar plots

Figs. 14 and 15 are versions of Fig. 5 for 10 and 25 configuration evaluations, respectively.

### G.4 Optimisation curves

We show the optimisation curves for NewsVendor, XGBoost and SVM 1220 in Fig. 16.

### G.5 Downstream performance

Table 4 shows that the downstream performance of SimpleOrdered is better than CTS. The reduction in standard error is calculated as $100(1 - s_{\mathrm{SO}}^{(i)}/s_{\mathrm{CTS}}^{(i)})$ for each task $i$ where $s_{\mathrm{CTS}}^{(i)}$ is the standard error of the performance across replications for CTS. $s_{\mathrm{SO}}^{(i)}$ is the same for SimpleOrdered.

The improvement in mean is calculated as $100(1 - m_{\mathrm{SO}}^{(i)}/m_{\mathrm{CTS}}^{(i)})$ for maximisation, and the same multiplied by -1 for minimisation. Here $m_{\mathrm{SO}}^{(i)}$ and $m_{\mathrm{CTS}}^{(i)}$ are the means across replications for SimpleOrdered and CTS, respectively. Note that we do not include the between-seed variation in our error estimate of the improvement in mean, only the between-task variation.

### G.6 Sampling locations

Fig. 17 shows sampling locations. As can be seen, SimpleOrdered and SimplePrevious combine very focused early exploitation with broad exploration later on.

### G.7 Ordering with different surrogate model: Density-Ratio Estimation

This section presents ablation results of combining the ordered approach in SimpleOrdered with BO using density-ratio estimation for the surrogate model (Tiao et al., 2021): DREOrdered. Note that the figures in the main paper were not replotted with these new results, although the metrics of the other methods can be impacted by the introduction of a new method. We also did not update the values in Appendix H to include these extra experiments.

We present results for DREOrdered in Figs. 18 and 19. As can be seen, the performance of DREOrdered is between that of SimplePrevious and SimplePreviousNoBO. This suggests that

Gaussian processes are better surrogate models for our problem. But comparing DREOrdered to standard transfer HPO we see that also with this surrogate model we outperform the non-ordered methods.

## H Compute budget

This summaries the compute costs for the results included in the paper (not including Appendix G.7).

The experiments were run on AWS Sagemaker, using ml.c5.18xlarge compute instances, which have 72 vCPUs, and 144 GiB memory. We ran a total of 65 experiments: 10 for NewsVendor, 10 for XGBoost, 10 for SVM 1220 (YAHPO) and 5 for each of the 7 remaining YAHPO combinations, so 35. We also trained and evaluated a total of 28000 XGBoost models for our XGBoost benchmark, also on AWS Sagemaker.

Collecting the XGBoost evaluations took a total of 79 hours and 46 minutes of compute time.

Collecting the experiment evaluations took a total of 231 hours and 39 minutes.

In total, we used 311 hours and 26 minutes of compute time for the results presented.

## I Broader impact

The intended impact is to make a subset of transfer HPO problems more efficient, with the positive impact of reducing energy consumption from training models. There is a risk that this will instead lead to the same computational budget being used and higher accuracies obtained, but that is just the lack of a benefit, not a harm in itself. However, we also show the benefit of doing hyperparameter optimisation for subsequent tasks as data set sizes increase. This could lead to more models being trained. By contributing simulation-based benchmarks the total energy consumption of future work should be reduced as the models do not need to be retrained. We see no ethical concerns with the data sets used.

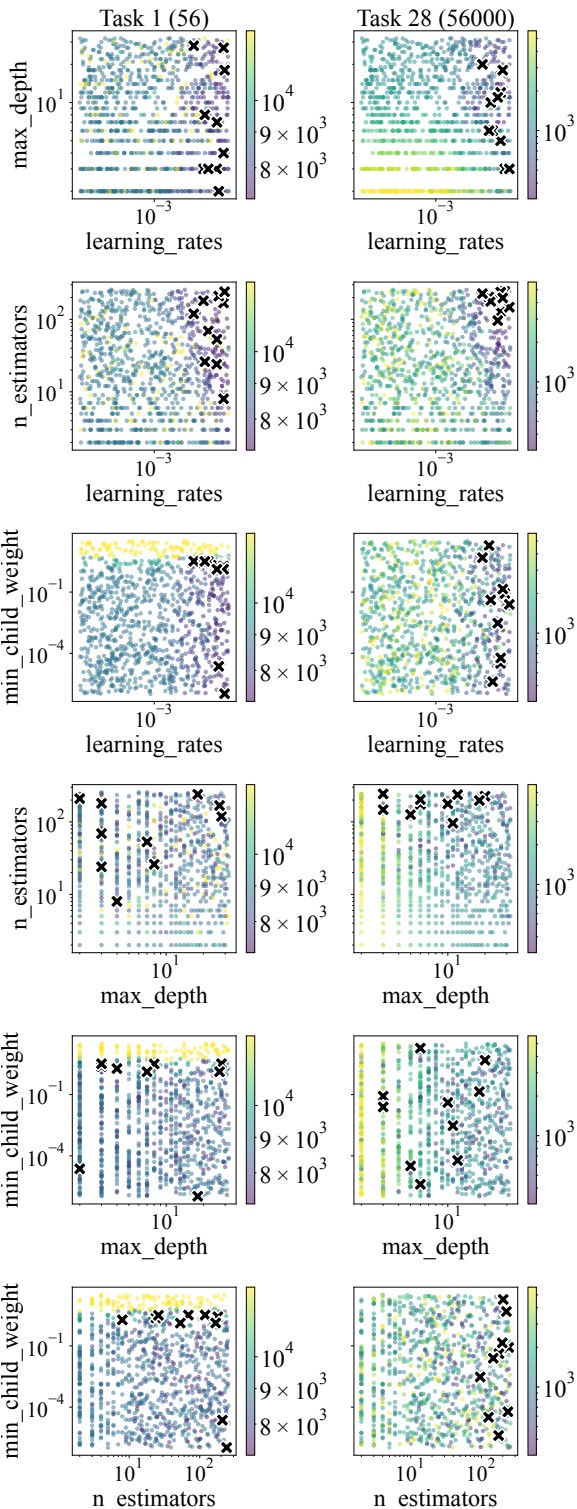

Figure 8: Hyperparameter landscape plots for all combinations of two hyperparameters of the XGBoost benchmark. Left: Task 1 with 56 data points in the training set. Right: Task 28 with 56000 data points. Black crosses indicate the top 10 configurations.

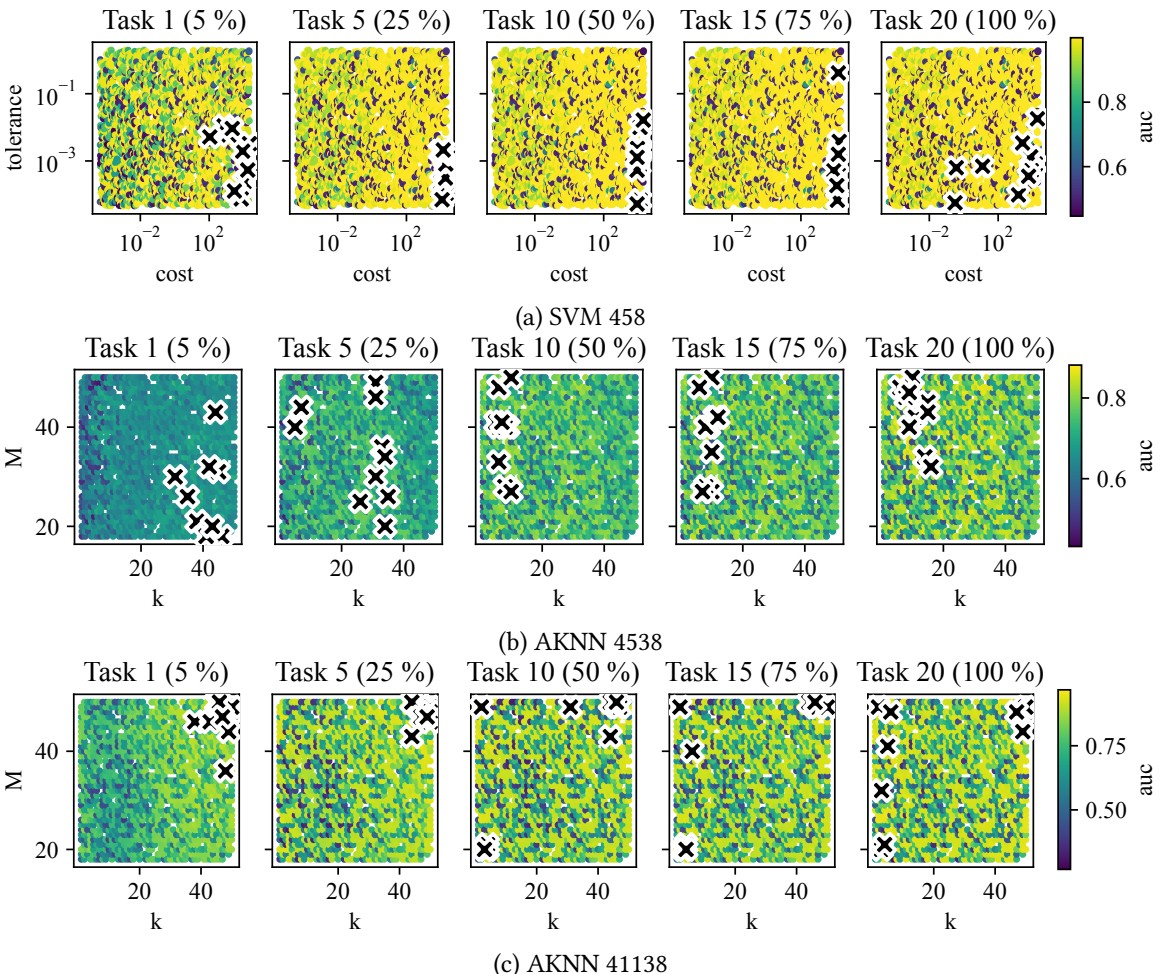

(a) SVM 458

(b) AKNN 4538

(c) AKNN 41138

Figure 9: Further plots of YAHPO hyperparameter landscapes showing ordered behaviour on SVM and AKNN models.

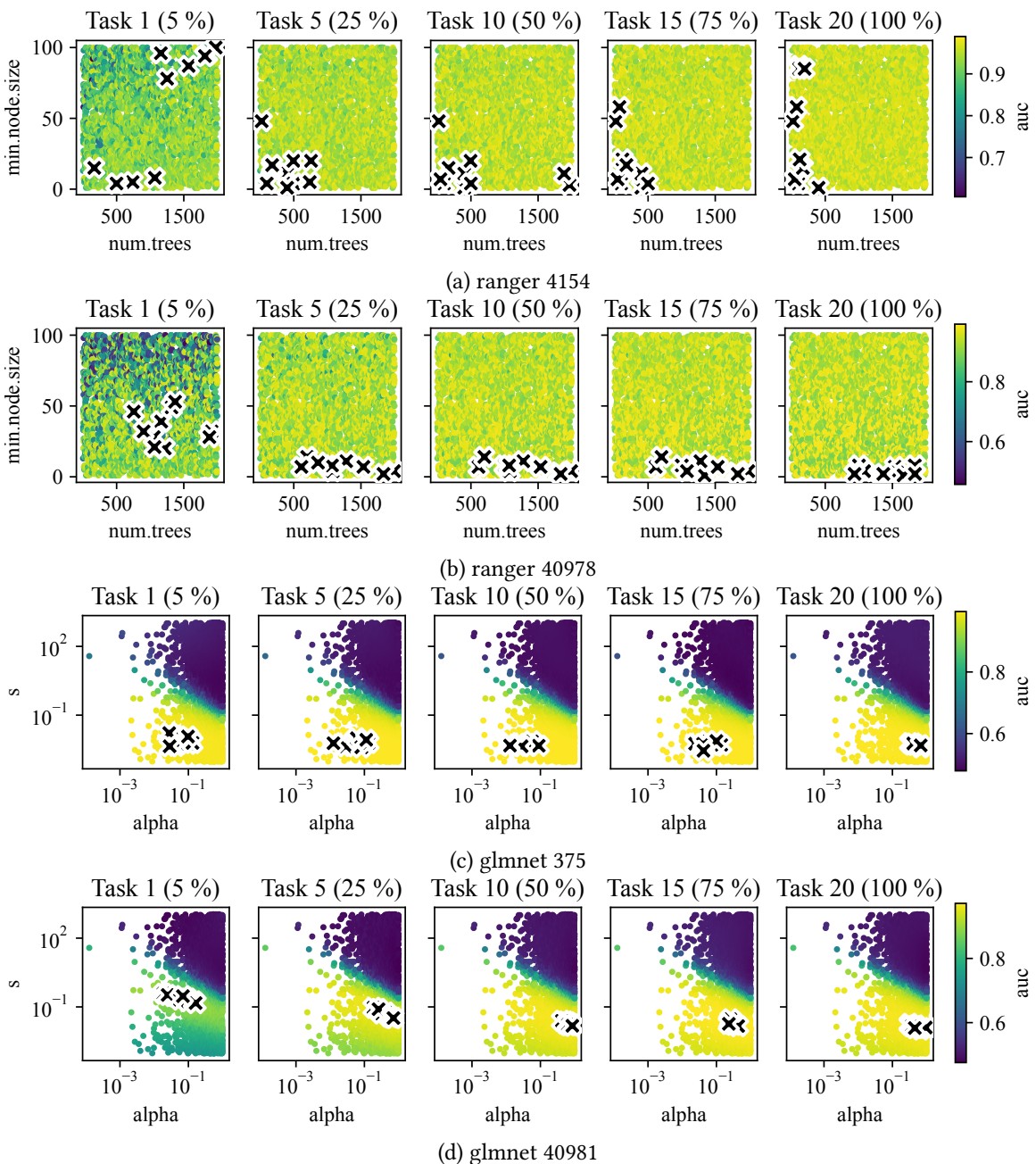

(a) ranger 4154

(b) ranger 40978

(c) glmnet 375

(d) glmnet 40981

Figure 10: Further plots of YAHPO hyperparameter landscapes showing ordered behaviour on ranger and glmnet models.

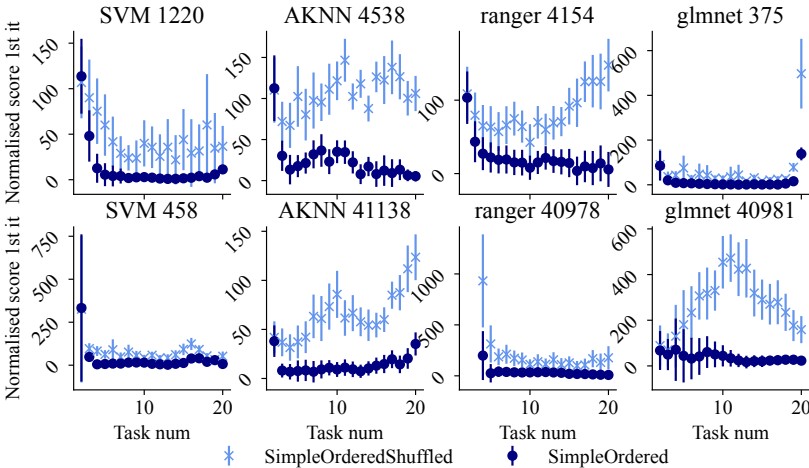

Figure 11: Mean normalised scores after first configuration evaluation for all eight YAHPO combinations considered. We see that for almost all the tasks SimpleOrdered performs better than its ablation SimpleOrderedShuffled.

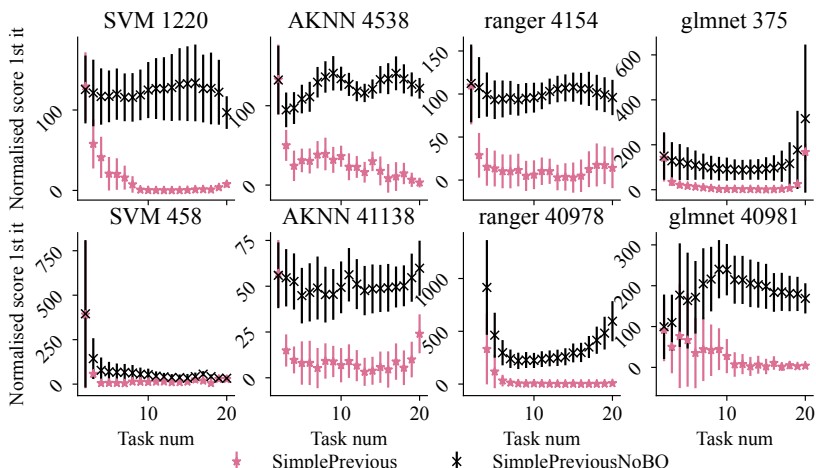

Figure 12: Mean normalised scores after first configuration evaluation for all eight YAHPO combinations considered. We see that for almost all the tasks SimplePrevious performs better than its ablation SimplePreviousNoBO.

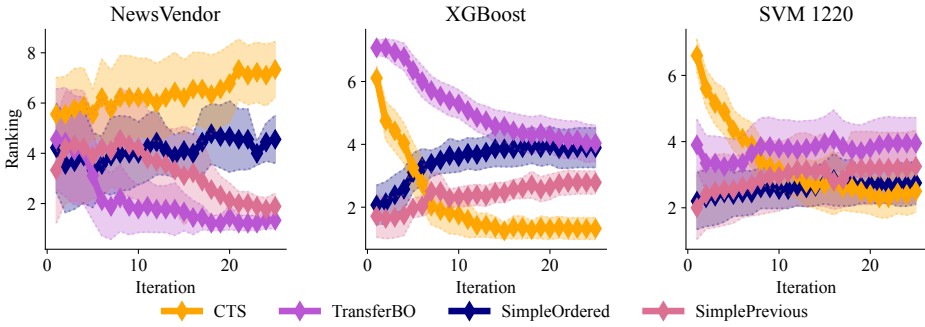

Figure 13: Mean rankings (± 2 standard error) as a function of configurations evaluated, averaged over tasks.

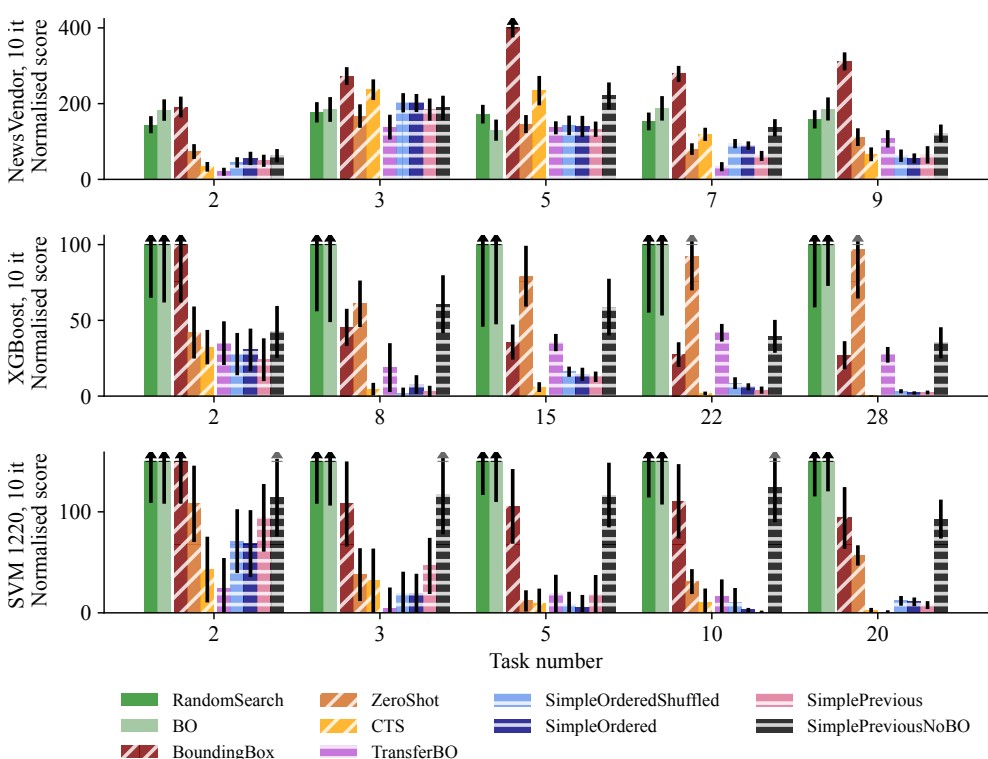

Figure 14: Mean normalised scores (+- 2 standard error) after the tenth configuration (lower is better). NewsVendor (top), XGBoost (middle) and SVM 1220 (bottom). Black arrows indicate that the mean was above the plotted range, grey arrows that the standard error range was above. Version of Fig. 5 for the tenth configuration.

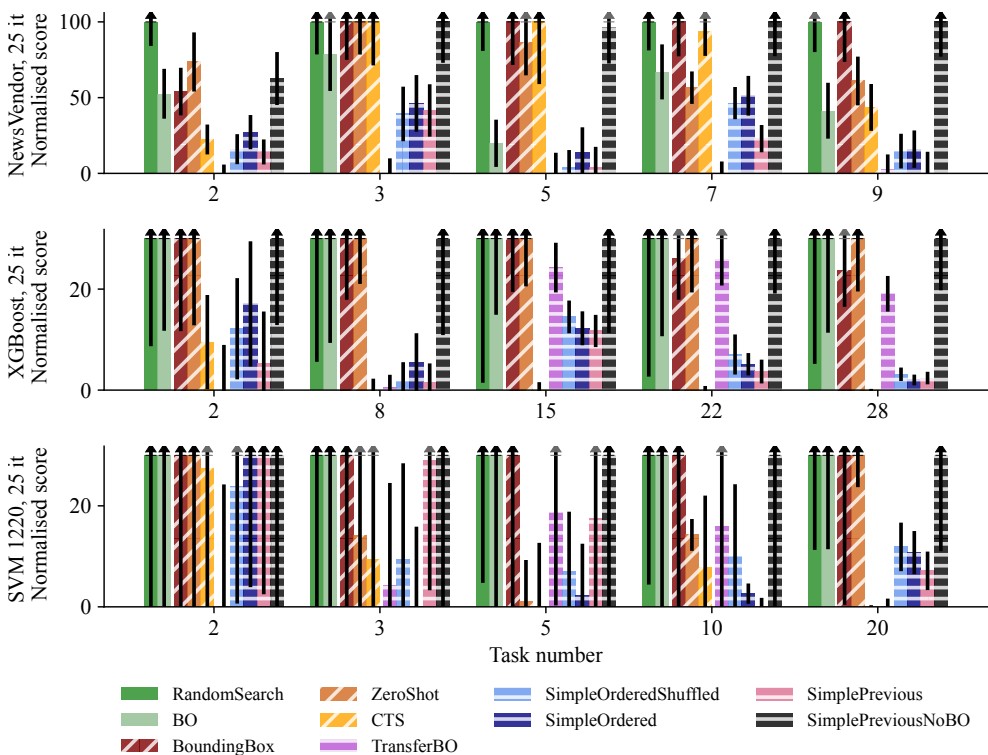

Figure 15: Mean normalised scores (+- 2 standard error) after the 25th configuration (lower is better). NewsVendor (top), XGBoost (middle) and SVM 1220 (bottom). Black arrows indicate that the mean was above the plotted range, grey arrows that the standard error range was above. Version of Fig. 5 for the 25th configuration.

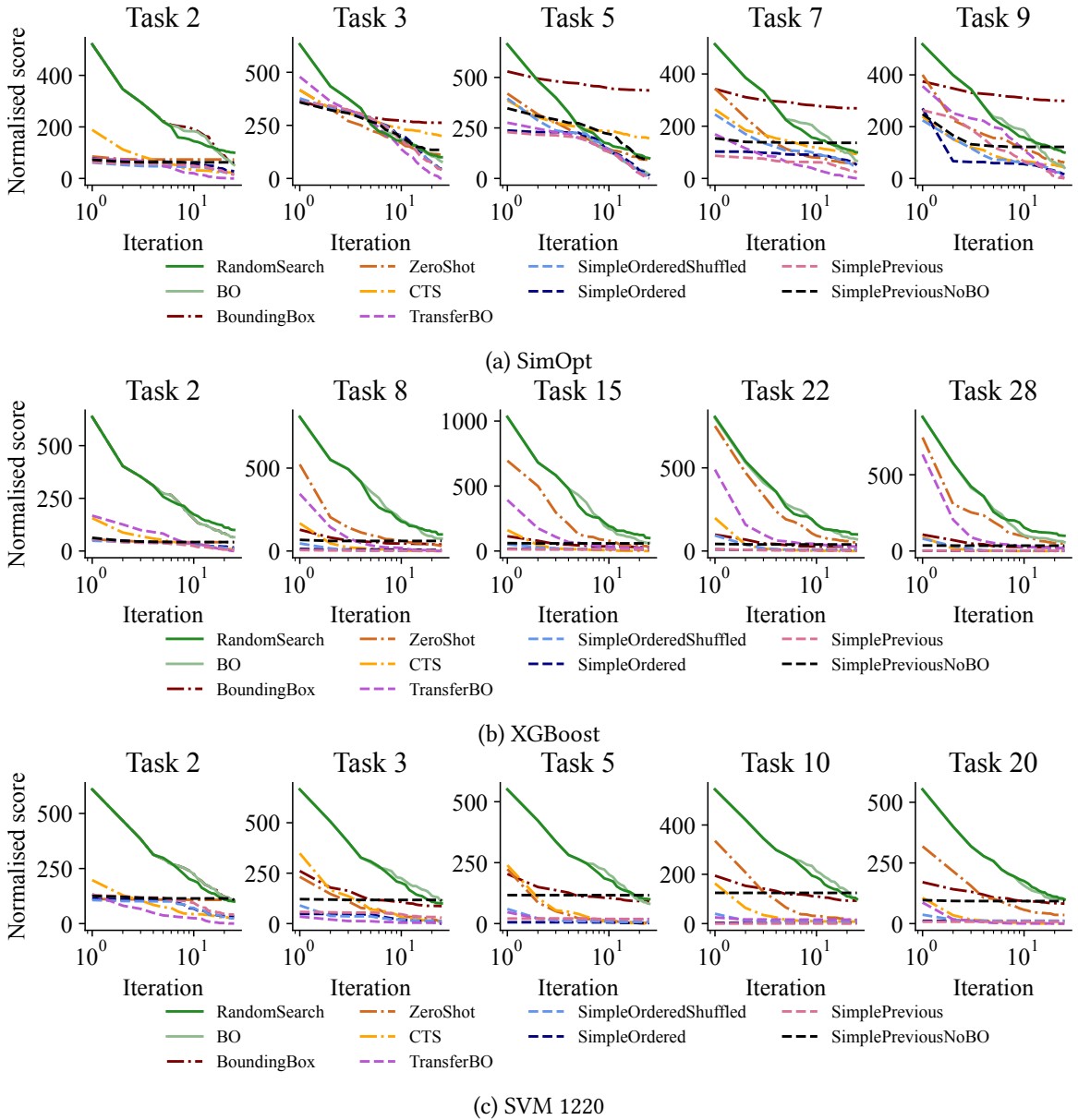

Figure 16: Companion plot to Fig. 5, showing the normalised score as a function of configurations evaluated for the same subset of tasks. SimpleOrdered starts off well, but does not improve much with more iterations.

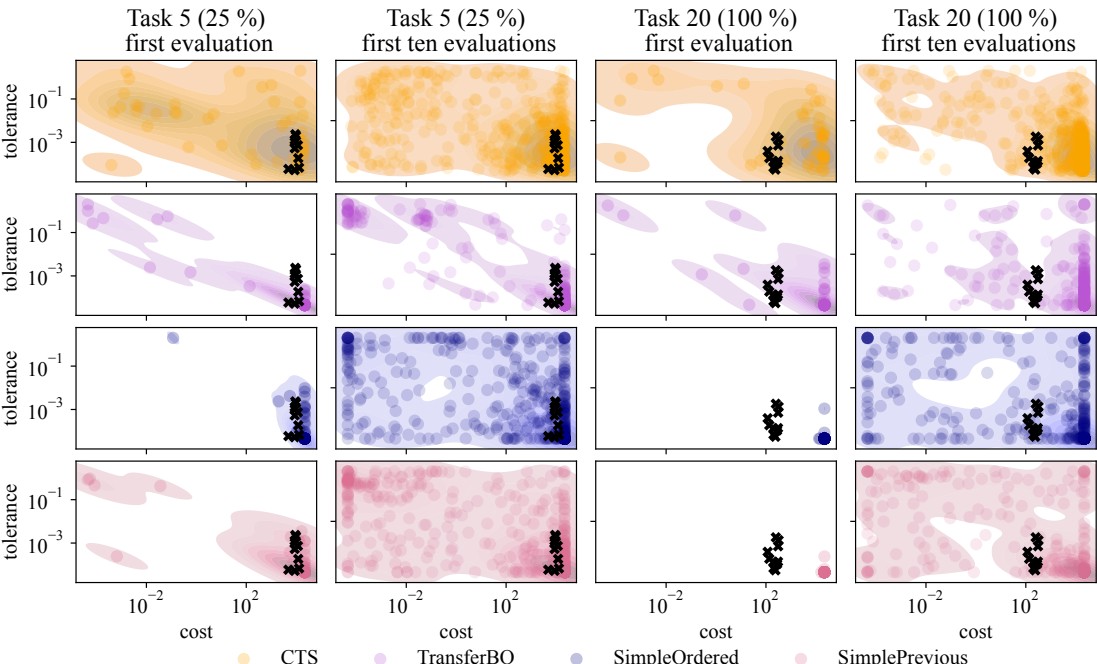

Figure 17: Sampling locations of the first and first ten evaluations for YAHPO SVM 1220 across the fifty replications. Black crosses indicate the top ten hyperparameter configurations. As can be seen, the SimpleOrdered and SimplePrevious samples are much more concentrated than the others for the first evaluation, and then is very explorative once the first five evaluations are done.

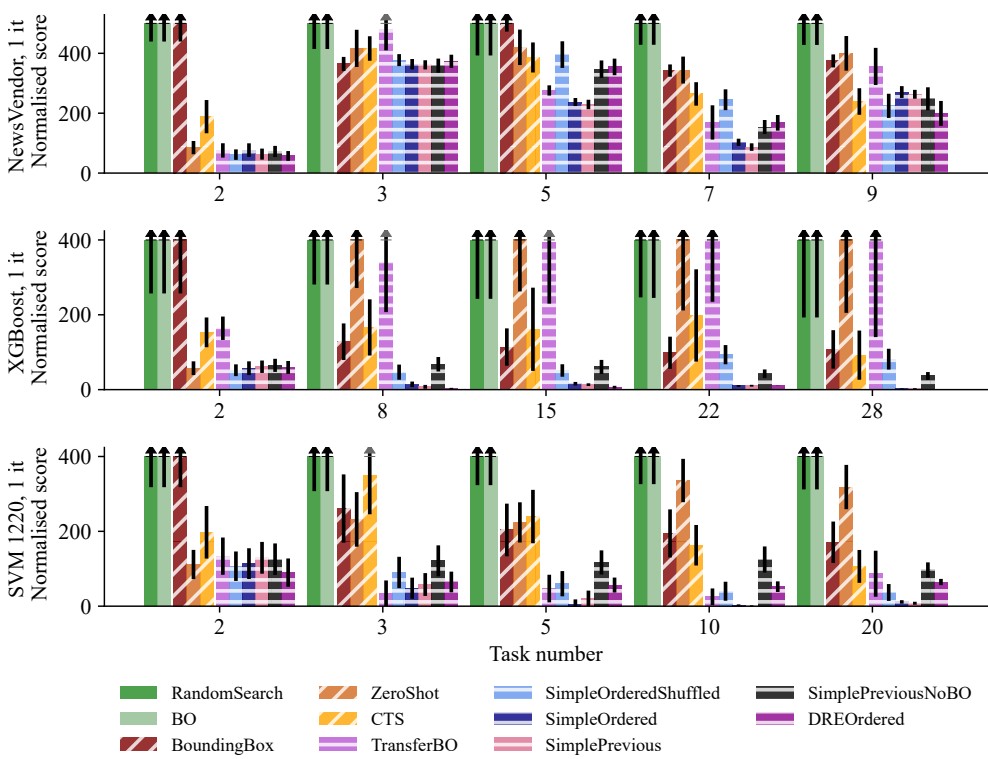

Figure 18: Mean normalised scores (+- 2 standard error) after the first configuration (lower is better). NewsVendor (top), XGBoost (middle) and SVM 1220 (bottom). Black arrows indicate that the mean was above the plotted range, grey arrows that the standard error range was above. Version of Fig. 5 with the addition of DREOrdered as a baseline.

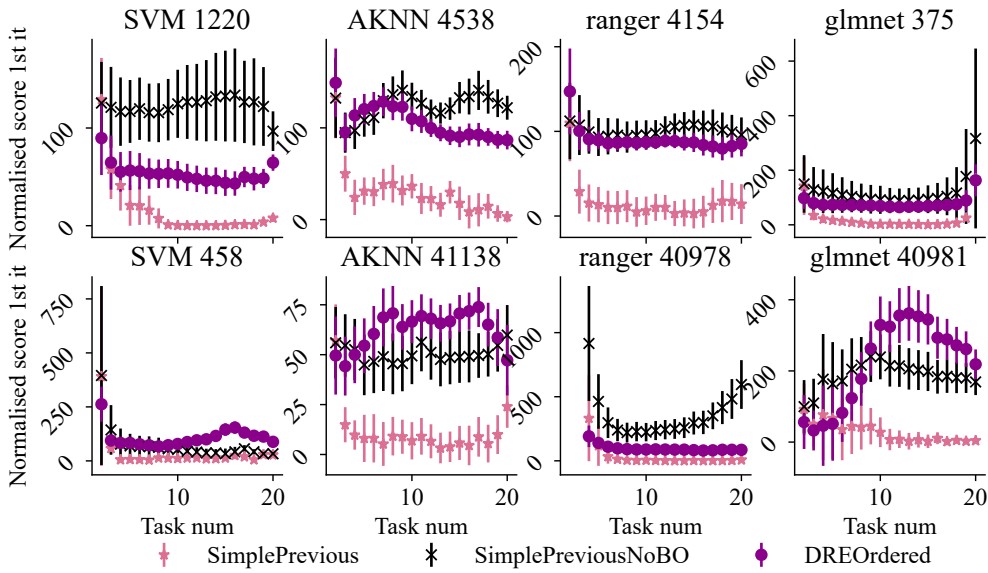

Figure 19: Comparison of new baseline DREOrdered with SimplePrevious and SimplePreviousNoBO.

