# OpenReview forum: "Obeying the Order: Introducing Ordered Transfer Hyperparameter Optimization"
_automl.cc/AutoML/2025/Methods_Track — AutoML 2025 Methods Track_

### Official Review · Reviewer_ciAA · 2025-04-25

**Comments To Authors:**

**Summary**: This paper introduces ordered transfer hyperparameter optimization (OTHPO), a new HPO task where the tasks follow a sequential order and each task is most correlated to those immediately before. In the end, the performance of OPTHPO tasks is evaluated on the last task in this sequence. The authors further propose 10 benchmarks with different ML models for this setup and show that simple methods, such as starting from the optimal configurations from the last task (or the last few tasks), could outperform the other sophisticated transfer learning HPO approaches.

**Strength**: Overall, this paper is well written and easy to follow.
* As stated by the authors, OTHPO problems are commonly applied in many scenarios. Given that the data distribution might shift over time, one might want to tune the hyperparameters of the target tasks more than once.
* Additionally, the proposed benchmarks make it easier for developers to evaluate new OTHPO problems.
* The provided two simple ordered transfer HPO methods, i.e., the simple ordered BO, which starts by evaluating top-performing hyperparameters from the most recent tasks, and the SimplePrevious, which only starts with the best configurations from the last tasks. Both settings show strong performance compared to all the other baselines.

**Weakness**: My biggest concern for this paper is that its ain't contribution is unclear. Do the authors only want to develop a new task and create a new benchmark for the task, or develop new approaches for solving the OTHPO tasks?

* In line 118, the authors claim that "but training and validation data sets change". However, only the training set changes in the XGBoost and YAHPO Benchmarks, while the validation set remains invariant.  Most of the tasks proposed in this paper might not reflect the general cases in the OTHPO setting. Therefore, if the main contribution of this work is to the benchmark, then the task types in this paper should be further explored and not limited to the benchmarks presented in this paper. Additionally, it is essential to show how each optimal hyperparameter configuration changes with the increasing tasks (for instance, with a parallel coordinate plot) to check if one can correctly construct the correlation between the configurations and the number of evaluated tasks.

* If the main contribution is the proposed methods. Then, the authors should provide a comprehensive analysis of the methods provided. For instance, the approach requires N hyperparameter configurations from the previous tasks. It is unclear how these N values affect the final performance. Additionally, this conclusion might also change as the number of evaluations for each task increases. Additionally, It is unclear if the user should use SimpleOrdered or SimplePrevious approaches given a new task. The authors only state that SimpleOrdered is slightly preferable on early tasks and SimplePrevious on later tasks (Linge 244). However, there is no further qualitative analysis on the criteria for selecting which methods to use in different scenarios.


**Questions**: From my understanding, if SimpleOrdered and SimplePrevious will provide the same results if only 1 task is evaluated (see section 5.2 and line 530). However, their performances are not the same in Figure 5 when there are only two tasks (for task NewsVendor and SVM 1220), is there any reason for that?

**Review Confidence:**

4

**Review Rating:**

6

---

### Official Review · Reviewer_mow9 · 2025-04-29

**Comments To Authors:**

(i) summary of the contributions
The paper introduces the problem of Ordered Transfer Hyperparameter Optimisation (OTHPO).
It differs from the known problem of Transfer Hyperparameter Optimisation (THPO), as it introduces relations between learned tasks (e.g. temporal relation in concept drift).
After the introduction, an overview of the state of the art for THPO is given. In Section 3, the problem is briefly defined in a formal way, before the used benchmarks are shown in Section 4. Section 5 outlines the conducted experiments, while Section 6 concludes the paper.

(ii) potential impact on the field of AutoML
A novel problem of OTHPO is introduced. This might spark further research and can potentially be applicable in the design of novel AutoML systems.

(iii) technical quality and correctness
3 different, relatively simple methods are given. Those outperfrom the given state of the art. The results are worthy to be presented to the  (auto)ML community.

(iv) clarity of the contributions
The definition of the novel problem and the three methods are clearly marked as own contributions. The formal definition of the problem, however, is somehow shallow.

(v) optional: things you note about reproducibility
Nothing noteworthy

(vi) potential ethical concerns
None

Final assessment:
Accept

Further Notes:
Please change the spelling in the title to optimization, as this spelling is much more popular (60 vs. 876 publications on dblp for HPO using either spelling). With the current spelling, the paper might be unnecessary hard to find, reducing its impact.
The context, inner and outer problem are not explained well in Section 3. Maybe, this could be better explained by a concrete example in the text.
In Figure 5. the arrows are hard to see. The striped bars next to each other are uneasy to look at.

**Review Confidence:**

3

**Review Rating:**

8

---

### Official Review · Reviewer_xXQM · 2025-04-29

**Comments To Authors:**

## Summary

The paper extends the HPO problem for sequence-trained prediction problems with the availability of more information (in this case it is treated as new data). Consequently, there is a dependency between sequential shuffles, which is called context in the paper. The HPO methods presented here take advantage of the dependence between successive tasks using the warm-start technique of Bayesian optimization. The results of these methods are contrasted with well-known HPO techniques using meta-learning, as these methods focus on information transfer between shuffles.
One of the authors' main contributions is to define a new optimization problem and show how it differs from standard transfer HPO, CL, and from multi-fidelity HPO.
In addition to defining the problem, the authors propose three benchmarks.

## Potential impact on the field of AutoML

The main contribution to the field of AutoML is the definition of a new optimization problem where successive tasks are connected through a shared context. However, in the presented approach, the context is quite limited as it solely reflects the availability of additional observations.

While the authors follow good practices by proposing a benchmark for testing future methods and providing three baselines that utilize the specifics of the new optimization problem, the benchmark's construction appears to be the work's weakest aspect (see Weakness).

### Strengths

- Definition of a special case of HPO where the following shuffles are similar to each other by context. The context in this case is due to the increasing information available. The strength of the paper is to show a new perspective of the optimization problem.
- Justifying why a problem is significantly different from previously analyzed problems: HPO with meta-learning, CL or multifidelity HPO
- A broad and thoughtful set of baselines was used
- Proposing 3 simple optimization methods that can serve as baselines in the future

### Weaknesses

-
- Definition of the problem: context as an increasing number of observations is primarily included.
- Reference to other possibilities for defining context is missing. No literature review on this issue.
- The description of the third benchmark is not clear to me, I do not see the connection to the problem.
- The benchmarks are rather limited - in fact, it is 1+2 + 1 different datasets only sampled to subsamples of datasets. I don't see the innovation here
- In the case of the XGBoost and YAHPO benchmark, authors do not take into account the potential change in the distributions of observations - when a subsample of observations is drawn, they are drawn directly from the entire available dataset which does not correspond to the case.
- Which leads to the question of how the problem relates to the phenomenon of data drift?
- Regarding increasing the training size in individual task, e.g. task 1 (5%) is a subset of task 5 (25%)? Such a way of creating tasks can distort the results a lot - then it would be good to do some variants of a larger dataset and see the stability of these results

### Minor weaknesses

- The example shown in Figure 1 is a rather well-known fact and follows directly from the costruction of the xgboost algorithm - the more data, the more estimators we need.
- In this work, the authors are limited to the case of task-is one dataset? Is it possible to transfer information between datasets

**Review Confidence:**

4

**Review Rating:**

3

---

### Meta-Review · Area_Chair_8oCW · 2025-05-08

**Recommendation:** Accept
**Confidence:** 4

**Metareview:**

Despite some concerns regarding the simplicity of the benchmark construction and the limited scope of the context definition, the reviewers agree that the problem formulation is both well-motivated and distinct from existing HPO settings. The proposed baseline methods are simple yet effective, outperfroming established techniques and demonstrating the value of modeling task order. The paper is clearly writte, reproducible with minimal effort, and provides resources likely to benefit future research. While further analysis and broader benchmark diversity would strengthen the work, the overall contribution is solid and likely impactful. Therefore, I recommend acceptance.